# OntoIMM: An Ontology for Product Intelligent Master Model

**Cong Yu [1], Fa-ping Zhang [1],\*, Shahid I. Butt [2]**  **, Yan Yan [1] and Wu Lv [3]**

[1]   School of Mechanical Engineering, Beijing Institute of Technology, Beijing 100081, China; yucong1984@163.com (C.Y.); yanyan331@bit.edu.cn (Y.Y.)

[2]   School of Mechanical & Manufacturing Engineering, National University of Sciences and Technology, Islamabad 44000, Pakistan; drshahid@smme.nust.edu.pk

[3]   China Ordnance Industry No.208 Research Institute, Beijing 102202, China; lvwu2005@126.com

\*   Correspondence: zfp_new@163.com; Tel.: +86-10-68912708-13

**Abstract:** Information organizing principle is one of the key issues of intelligent master model (IMM), which is an enhancement of the master model (MM) based on KBE (knowledge-based engineering). Despite the fact that the core product model (CPM) has been confirmed to be an organizing mechanism for product master model, the key issue of supporting the information organizing for IMM is not yet well addressed, mainly due to the following two reasons; (1) lack of representation of complete information and knowledge with regard to product and process, including the know-why, know-how, and know-what information and knowledge, and (2) lack of semantic richness. Therefore, a multiaspect extension to CPM was first defined, and then an ontology was constructed to represent the information and design knowledge. The extension refers to adding a design process model, context model, product control structure model, and design rationale model to CPM concerning the enhancement of master model, which is to comprehensively represent the reason, process, and result information and knowledge of the product. The ontology construction refers to representing the concepts, relationships among these concepts and consistency rules of IMM information structure. Finally, an example of barrel design and analysis process is illustrated to verify the effectiveness of proposed method.

**Keywords:** intelligent master model; information organizing principle; core product model; ontology; product design

---

## 1. Introduction

With the development of design complexity on requirement, function, and structure of modern engineering product, the development process has become more and more complex. The complexity is increasingly controlled by distributed groups with many phases and function views of product development process, and product development across and within the companies will almost invariably take place within a heterogeneous software environment [1]. Furthermore, engineering product development is a knowledge-intensive process [2,3]. Designers are no longer merely exchanging geometry data and information, but also product and design process knowledge, which requires the acquisition, representation, management, and application of various kinds of design knowledge [4]. As a result, there is a greater need for the integrated, knowledge-intensive and multiview information support mechanism [5].

The product master model (MM), a framework first presented by Hoffman et al. [6] to keep consistent associations between CAD and downstream applications, has proven to be an effective solution to address the aforementioned challenges. It acts as the persistent representation of the

product, maintaining consistency and integrity of product data across the different functional views. Master model is enhanced by fusing with KBE, which extends these efforts with the capture of designer's knowledge to create intelligent master model (IMM) [7,8]. IMM represents completely product information and design knowledge about the reason and the process concerning product and its development process. It integrates product definition with high fidelity and interfaces of design and analysis systems or codes such as CAD, finite element analysis (FEA), and other calculation codes, so as to serve the product development process intelligently. Consequently, the introduction of IMM process has been a key to enabling true concurrent design [9].

For the multiview information architecture embodied in the MM/IMM, there are two key and basic issues [10]: what should be the organizing principle for structuring the information and what principles should govern the implementation. In this article, we mainly concentrate on constructing the information organizing mechanism for intelligent master model based on previous efforts for organizing multiaspect information for MM.

Three approaches of structuring information have been adopted to serve as the organizing principle for the master model, which are: net shape [6], feature [11], and CPM [12]. The net shape mechanism, as the means of connecting non-geometric functional information to shape elements of product, provides the function that it accommodates aggregate or composite shapes as well as the primitive shape elements. However, the biggest shortcoming is that the multiaspect information architecture should be universally applicable, even in the absence of shape information. Similarly, to use features [11] as the organizing principle has two similar issues: non-geometry information, such as function information, may have to be associated with semantic entities more abstract than features, and functional information needs to be organized even in the absence of assigned features. The CPM [12] and its modifications/extensions have been confirmed to be an effective method serving as the organizing mechanism for a range of design-related information structures. However, the CPM is restricted to a typical set of attributes, which primarily due to a category of static know-what information. Domain-specific attributes and object-specific attributes (e.g., function, behavior, or form) are excluded in the representation intentionally [13]. Obviously, the CPM also needs to be further expanded and applied for organizing information and design knowledge about product and process completely, including know-why and know-how.

Despite the fact that CPM and its extensions have been developed to support acquiring and sharing product information in heterogeneous software environment for engineering development, the challenges of supporting the complete information organizing for intelligent master model are not yet well addressed, mainly, for the following two reasons.

(1) Lack of representation of know-how and know-why information and knowledge about product and its development process that facilitates design decision-making and the completeness of know-what information. CPM mainly focus on capturing know-what information of an artifact, such as requirement, function, behavior, and form, while lacking the organization of know-why and know-how knowledge. However, the acquisition and fusion of design knowledge is the key of the main enhancement of rendering master model to intelligent master model.

(2) Lack of semantic richness of representing product-related information and enabling multiview engineering analysis that supports different stakeholder viewpoints and heterogeneous systems in collaborative environments across the product life cycle. Semantic richness of the representation of product/process information is critical for information exchanging, sharing and interoperating. The semantic representation of CPM has been taken into account in some studies [14]. However, as mentioned in the above stated shortcoming 1, the CPM mainly focuses on capturing know-what product information. The extended know-why and know-how information and design knowledge also requires semantic representation.

To address the aforementioned needs, we define an extension to CPM as the information organizing mechanism of IMM and construct an ontology to represent it. First, four submodels are added to CPM

to comprehensively describe the reason, process, and result information and knowledge. Second, the concepts, relationships, and consistency rules of IMM are modeled in an ontology.

The remainder of this article is structured as follows. The foundations of this paper, MM/IMM, CPM, and ontology, are briefly reviewed in Section 2. Multi-perspective extensions to CPM are described in Section 3. The formal modeling of concepts, relationships, and consistency rules is presented in Section 4. Section 5 is an example of barrel design and analysis. Section 6 covers the concluding remarks.

## 2. Related Works

### 2.1. Product Master Model and Intelligent Master Model

Engineering product development refers to integrating various process models, including geometric model, dynamic model, quality analysis model, and finite element model and so on. To achieve the integration and interoperability, a common model must exist to provide different views for different analysis domains [15]. The master model approach, which automatically links the different functional analysis view models, is one way to address this challenge [7,8,16]. Master model [6] is a single product definition to enable rapid design and analysis iterations. It provides all the information necessary for design and analysis, and then accesses results information to update itself. In this way, the product information is updated during the multidisciplinary design optimization (MDO) process [17].

Many researchers have carried on deep research on MM theory and obtained meaningful applications. Yuan et al. [18] argued that MM is a software-based central database which deposits all kinds information of geometry and non-geometry concerning product and process. Lee [19] proposed a new CAD/CAE integrated strategy based on MM to create the required CAD and CAE models. Rocco et al. [20] described the development of a MM concept of the DEMO vacuum vessel (VV), and in order to simplify future finite element (FE) analyses, a surface modeling technique was adopted and efficiently linked to the FE (Finite Element) code.

With the fusion of KBE and product control structure (PCS), linked model environment (LME), MM is rendered as an intelligent master model (IMM). IMM first proposed by the FIPER project [7,8] under the sponsorship of the National Institute for Standards and Technology (NIST). In addition to know-what information of design, IMM captures the intent behind the product design by representing know-why and know-how knowledge concerning product and its development process. The geometric definition is only one view (aspect) of the product information associated with the total product definition. It also contains part dependencies, geometric and non-geometric attributes, quality and cost constraints, and manufacturing producibility.

In order to enhance MM to IMM based on KBE, many efforts have been made. Cedar et al. [9] suggested that an extension to IMM process is the use of KBE technique, which allows rapid changes in engine configuration and automation of the definition of complex engine components. To address the intelligence in process design and the integration of knowledge and design methods, Wang et al. [21] proposed a process design IMM based on KBE to integrate design process and relevant technologies. Rocca et al. [22] reported that KBE has been exploited to develop a flexible design system and to integrate a heterogeneous set of distributed discipline-specific design and analysis tools into a modular design framework. The core units of the framework contain multimodel generator (MMG), which can generate many aircraft component configurations based on high level primitives (HLPs). Marcus et al. [17,23–25] presented a knowledge-based master model approach (KBMM) to integrate design and analysis models. The master model approach promotes the existence of a single governing version of the product definition as well as operating scenarios. To provide more flexibility in geometry change and linked analysis model, KBE can be used to control the MM. Compared to traditional parametric CAD systems, and since KBE can enable more flexible geometry configuration, they argued it is possible to further enhance the master model ideas by using the capabilities of KBE software within a CAD system. However, the research efforts mainly concentrate on the design automation

based on KBE, especially for some geometry modeling, lacking the support of other intelligent design activities, such as system level controlling, reasoning and judging, design decision-making.

Another key issue concerning multiaspect information architecture is the information organizing mechanism. There are three approaches of structuring information have been adopted to serve as the organizing principle for master model, including net shape [6], feature-based [11], and CPM [12]. Net shape mechanism offers the function to accommodate aggregate or composite shapes as well as the primitive shape elements. However, even in the absence of shape information, the multiaspect information architecture should be universally applicable. Similarly, feature-based approach [11] has two similar issues: non-geometry information may have to be associated with semantic entities more abstract than features, and functional information needs to be organized even in the absence of assigned features. Incontrast, CPM [12] and its extensions have been confirmed to be effective methods serving as the organizing mechanism for a range of design-related information structures. However, CPM is limited to a typical set of attributes. The representation intentionally excludes domain-specific attributes (e.g., attributes of mechanical) or object-specific attributes (e.g., attributes of function, behavior, or form) [13]. Obviously, the confirmed CPM is not adequate to serve as the information organizing mechanism for IMM.

As multiaspect information architecture for intelligent product development systems, IMM approach still needs to be studied deeply and systematically. The research includes information organizing mechanism and implementation principles; the former is the mainly concentration of this paper.

## 2.2. Core Product Model

The core product model (CPM) was first presented by NIST [1]. The primary objective was to provide a common data model as a base-level data representation for a multilevel design information flow model [26]. It presents a generic product representation architecture for whole product development process [27]. CPM is not tied to any vendor software. It is open, expandable, simple, generic, nonproprietary, and independent of any one product development process. CPM enables capturing the engineering information which is most commonly exchanged and shared in product development activities. It concentrates on the representation of artifact, which is used to describe a distinct entity in a product, whether that entity is a product, component, part, assembly, or subassembly. The description of an artifact refers to requirement, function, behavior, form, geometry, and material; functional and physical decompositions; and relationships among these concepts.

An entity hierarchy of the class Artifact is used to represent a product [14]. Artifact is an aggregation of Function, Behavior, and Form. Function represents what the artifact is supposed to do. Behavior represents how the artifact implements its function. Form represents the proposed design solution for the design problem specified by product function. The class Form is the aggregation of Geometry and Material, the former is the spatial description of the artifact, while the latter is the internal composition.

In addition, there are three utility classes—Information, ProcessInformation, and Rationale. ProcessInformation is an attribute set which is related to product development process, such as process state and schedule, input and output, process data version designation or other process descriptors. The class Rationale describes explanatory information on the reasons for or justifications of a particular decision concerning the artifact.

Some extensions to CPM have been applied to other concerns. In order to provide a standard representation and exchange protocol for assembly, Rachuri et al. [28,29] provided an open assembly model (OAM) which is an object-oriented definition of an assembly based on the extension of CPM. Sun et al. [30] defined a design domain ontology based on CPM. Wang et al. [31] proposed the product family evolution model (PFEM), which is an extension of CPM to represent the evolution rationale of product families and the rationale of change. Moreover, Lee et al. [32] proposed a multilevel product modeling framework to enable stakeholders to define product models and relate them to

physical or simulated instances. Biswas et al. [33] suggested extending CPM to components with continuously varying material properties. Shooter et al. [34,35] presented a design information flow model to eventually support a semantics-based approach for developing information exchange standards. Xu et al. [36] described a modeling framework to support conceptual design of multiple interaction-state mechatronic devices using state transition diagrams based on CPM. Zha et al. [12,37] presented a feature-based approach to the codesign of hardware and software in embedded systems through extending of CPM. The physical artifacts defined in CPM are translated into informational artifacts (e.g., hardware, software, business processes, organizations, and plans), through some modifications/extensions. In addition, in order to support tighter integration of spatial design and functional analysis, Fenves et al. [10] proposed a conceptual data architecture. In the architecture, CPM serves as the information organizing mechanism of MM which generates discipline-specific functional models.

The CPM and its modifications/extensions, serving as the organizing mechanism for a range of design-related information structures, have been confirmed to be an effective method. However, the CPM is restricted to a typical set of attributes which require to capture generic product information and to create relationships among them to make the representation as robust as possible. The representation intentionally excludes domain-specific attributes or object-specific attributes [13]. Obviously, the CPM also needs to be further expanded and applied for specific domains. For example, the core model could be extended to encompass activity and actor classes for the extension of MM to encompass management related data [10], extension for information completeness and comprehensiveness, and model implementation based on Ontology Web Language (OWL). Additional issues about product information modeling may include configuration and version control [32].

### 2.3. Ontology

Originally, the term ontology derives from philosophy. It is the description of the existence of beings in the world. In computer science, artificial intelligence (AI) researchers adopted the term ontology to describe what can be represented of the world in a program [38]. It is usually defined as an "explicit, formal specification of a shared conceptualization" [39]. Because of advantages such as implementing interoperability, integrating different applications, sharing information, and reusing knowledge [40], ontologies have been widely used to serve as generic information models in many systems or domains and facilitate semantic interoperability [41]. Lee et al. [32] presented an ontology-based multilevel product modeling framework to address the need of semantic richness from different stakeholders across the product life cycle. Panetto et al. [42] reported an approach based on ontology for facilitating applications interoperability in a manufacturing environment. Based on an ontology, Barbu et al. [14] created OntoSTEP which can transform digital models with geometric information into semantically rich models. Vegetti et al. [43] presented an ontology called PRoduct ONTOlogy (PRONTO) for modeling product data, which intends to provide a consensual knowledge model in product modeling domain. In order to enable the semantic interoperability across different application domains, Patil et al. [27] proposed an ontology-based framework by using product semantic representation language (PSRL) to represent product information related. The PSRL is a basis for a formal representation of product information [10]. In addition, many ontologies have been created, such as domain ontologies, generic ontologies [44], application ontologies [45], representation ontologies [46], and method and task ontologies [47].

The two most widely-used ontology modeling paradigms are OWL and Frame [48]. The Frame Ontology [46], which allows expressing knowledge in a frame-based or object-oriented way, defines concepts such as frames, slots, and slot constraints. The ontology is suitable for modeling IMM because that it is based on a closed-world assumption where everything is prohibited until it is permitted.

### 3. Extensions to Core Product Model for Intelligent Master Model

*3.1. Main Diagram of Intelligent Master Model*

The organizing principle of IMM is an extension of CPM based on KBE. The main diagram of intelligent master model is shown in Figure 1.

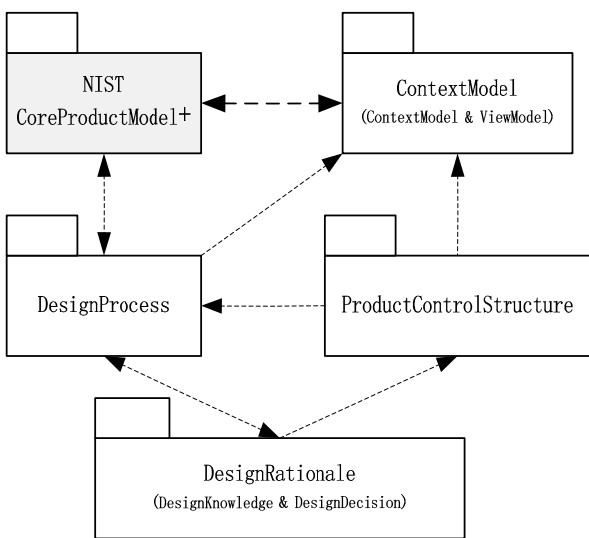

**Figure 1.** Main diagram of intelligent master model.

Figure 1 illustrates the five submodels of IMM, namely, CoreProductModel+, design process, product control structure, design rationale, context model, are defined in the packages CoreProductModel+, DesignProcess, ProductControlStructure, DesignRationale, and ContextModel, respectively. The dependency relationships (represented by dashed arrows) show that there exist some correlations among classes defined in different packages.

The information organizing mechanism of IMM first reuses classes and relationships defined in CPM and some extensions, so as to organize complete product information such as requirement, function, behavior, form, geometry, material, parameter, and constraint. In addition, in order to represent the knowledge about the reason and the process of product and its development process, the class ProcessInformation, defined in CPM, is extended to be design process model; Rationale is extended to design knowledge and design decision model; andthe product control structure model and context model are integrated to design process model to represent the top-down design method and multidomain analysis model. The following diagrams and paragraphs introduce more detailed descriptions of these packages.

*3.2. CoreProductModel+*

CPM is the core of information organizing. The extension first reuses the classes and relationships defined in CPM, such as Artifact, Requirement, Specification, Function, Behavior, Form, Geometry, Material, Feature, Port, ProcessInformation, and Rationale. The extension also reuses the classes and relationships defined in other extensions, including the term Assembly and AssemblyRelationship defined in OAM [28]. Furthermore, two concepts—Parameter and ParameterRelationship—are added to CPM for representing the parameter and the relationships among parameters. Parameter is a specialization of Information, and exists extensively in all kinds of product information, such as requirement, function, behavior, geometry, and material. The relationship among parameters is specialized into four relationships, including function mapping, model mapping, knowledge mapping, and geometry mapping. Function mapping expresses mathematical relationships among parameters or the relationships between parameters and non-parameters. Model mapping is the relationship

between model and its parameters. Knowledge mapping is a kind of knowledge representing method, such as generative rules and description logic, to construct the relationship that cannot be modeled by mathematical formula. Geometry mapping is a kind of experiences summary of many complex engineering problems which cannot be abstracted into concrete theories, for instance, the P–L curve for barrel design. In all, the reuse and modifications/extensions have been strengthened the traditional core model to be an enhanced core product model, called CoreProductModel+.

### 3.3. Product Design Process Model

Engineering product development is a knowledge-associated process which is not only required to process a large amount of information and knowledge, but also generates a vast amount of intermediate and result knowledge, in the form of documents, CAD models, CAE results, and input/output parameters. Therefore, the description for design process requires defining the activities and relevant process elements, such as input/output, conditions and tools, to support the management of design process and process data. The main concepts and the relations of design process model are shown in Figure 2. The following concepts are needed to model the design process model.

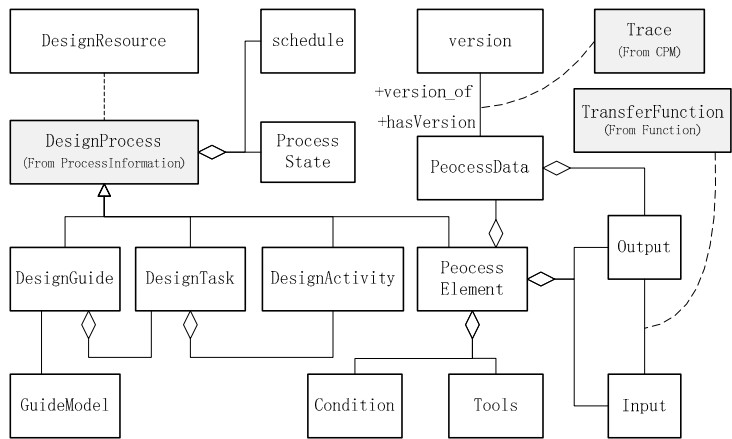

**Figure 2.** The main concepts of design process mode.

DesignProcess model is the extension (or specialization) of the class ProcessInformation of CPM. The elements of DesignProcess model includes design guide (supported by guide model), design task, design activity and process elements, schedules, process states, and basis attributes. Product information is generated and fulfilled gradually in design process, with the support of design resources such as design knowledge, design decision-making methods, design tools, and participants.

DesignGuide consists of logic and interrelated design tasks which are composed of specific design activities. DesignGuide is supported by a guide model (called GuideModel), through which the design requirements, engineering constraints, design knowledge, tools, and programs are integrated. Even inexperienced designers can quickly grasp design steps and the knowledge involved in design.

DesignTask model is adopted to describe task information, such as task goal, task description, and input/output data, and design team.

ProcessElement represents the execution conditions, input/output, design, and analysis tools. Output data, the result of process execution, is a kind of process data (defined as ProcessData class). ProcessData has a required attribute of Version; a new data version is obtained for each process execution. The relationship "version_of" is an attribute of Trace defined in CPM.

### 3.4. Product Control Structure Model

Product control structure model (PCS) allows engineers to layout the product configuration and controls engineering changes in a top-down format. What-if analysis at different levels is facilitated by allowing the designers to evaluate alternate configurations or to make parametric changes [7,8]. PCS is

intelligently modified by KBE (especially knowledge rules) to drive the change of product structure and configurations, and thereby intelligently scales a complete product or components of the product.

The hierarchical decomposition of product into its systems, subsystems, and components of PCS are represented by multilayer skeleton which is the determined factor of product configuration. It controls the model shape, feature dimensions and part locations, and assembly relationships through high-level product attributes and key datum planes, axes, points and parameters. Once subsystems establish and reference the top-level datums, each subsystem can be designed independently in a distributed format and later be assembled automatically.

Within PCS, the morphological features of components may be represented by preliminary, simplified geometry or just datums, the topological relationships between components may be controlled by geometry interfaces. Geometry interface is a kind of interface to represent the spatial location, orientation and assembly relationship of parts and components. Interface templates may be defined to represent the geometry interface information (including category, geometry description, interface parameters, and constraints) and reused, since the interfaces tend to be stable for the similar engineering product. The main concepts of PCS are shown in Figure 3.

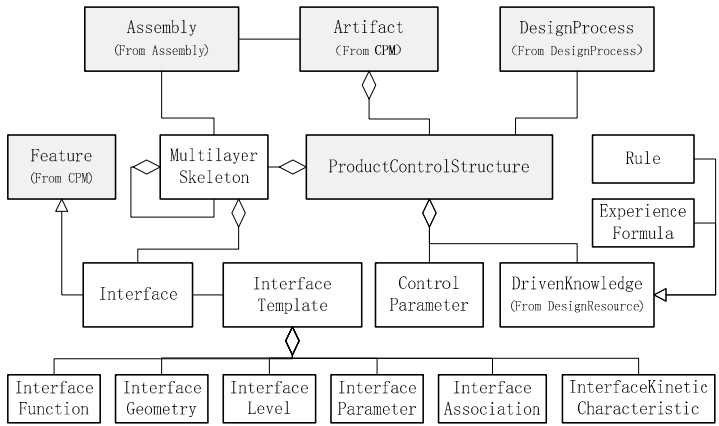

**Figure 3.** The main concepts and relationships of product control structure (PCS).

The following concepts are needed for modeling the PCS model.

ProductControlStructure is designated for controlling product structure in morphological or topological ways based on knowledge rules. Besides the basis attributes of name, type and information inherited from the class Artifact, it also includes multilayer skeleton, control parameter, and driven knowledge.

MultilayerSkeleton is the determined factor of product configuration, which controls the model shape, feature dimensions and part locations, and assembly relationships, through high-level product attributes and key datum planes, axes, points, and parameters.

ControlParameter is a parameter set of configuration parameters and sizes which drive the generation or update of product model. Within multilayer skeleton, the top skeleton is a bridge connecting to each other and storing the position relation and global parameters of each part. These global parameters control the configuration and size of the subordinate parts. The lower skeleton replicates these parameters and coordinates the skeleton information of its own to determine the overall configuration size of the current component, and controls the position relation of the subordinate parts, and passes down the global parameters.

InterfaceTemplate represents the interface relationships between levels and inner level of product assembly can be instantiated as an interface of multilayer skeleton. A PCS may include several interfaces as it is an assembly structure. For example, gun PCS may have interior ballistic interface, gun-bullet interface, muzzle interface, barrel interface. These properties of InterfaceNO, InterfaceFunctionStructure, InterfaceGeometry, InterfaceLevel, Interfacekinetic, InterfaceParameter, and InterfaceAssociation may be used to define an InterfaceTemplate.

DrivenKnowledge, especially design rules and experience knowledge, are used to intelligently modify the PCS in morphological or topological ways. It drives changes to parameters that define cross-sections and features and thereby intelligently scale a complete product or components of the product. Driven knowledge is a specialization of design knowledge. PCS is a specialization of artifact defined in CPM, and connects with design process. The multilayer skeleton includes assembly corresponding to the class Assembly defined in OAM.

### 3.5. Multi-domain Context Model and View Model

Engineering product development is an integrated multidisciplinary process. Each discipline uses different tools to design and analyze based on the discipline model with different abstract product definition. The domain-specific models are the information extracted and reconfigured from intelligent master model, such as geometry definition, structure parameters, behavior, and performance parameters, which makes the IMM to generate discipline-specific model in all levels. The terms ContextModel and ViewModel are used to record the context attributes of discipline and the product data view concerning specific design tasks and discipline context.

ContextModel describes the context information such as discipline, data type, the requirements, conditions and required operations for generating domain-specific model. These terms of Domain, Intension, Fidelity, Level, Design stage, and Operation are used to define context model.

ViewModel acquires the analysis data for specific analysis based on the task discipline information. The analysis data includes function, behavior, structure, material, load, and constraints which are extracted from CPM+.

The context model and view model is shown in Figure 4.

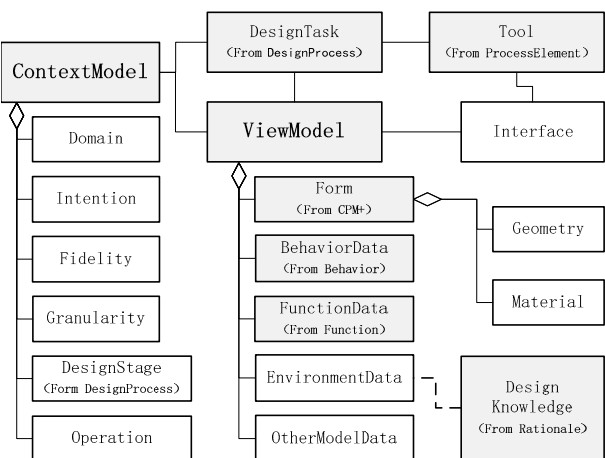

**Figure 4.** Multidomain context model.

As illustrated in Figure 4, the context model and view model are in relation to design process model. Hence, the generation of context model and view model should be according to specific design task, and the view data of view model is transformed to design analysis tools which are process elements in design process model. Depending on design requirements, the models generated for functional analysis or simulations have different fidelity levels. Rules, macro commands, scripts, or API commands are used to generate and govern view models. As product components are designed for system-level analysis, the IMM governs the changes at system level, subsystem level, component level, or part level. Variables and parameters linked in all view models are updated by the IMM automatically. In this way, other functional analyses can be conducted, and product definition changes can be guided accordingly.

### 3.6. Design Knowledge and Design Decision Model

Engineering product design is increasingly recognized as a decision-making process [49,50]. Product information is constantly completed based on design decision during product development process. However, the availability of necessary information influences quality decision-making. In general, engineers make design decisions or trade-off analysis based on their own experience and knowledge of what he can find [51]. In order to acquire the reason and the process, the concept Rationale defined in CPM is specialized into design rationale which is further specialized into two specific types: design knowledge and design decision.

DesignRationale, the specialization of Rationale defined in CPM, represents the knowledge on the reasons for or justifications of a particular decision in the product development process.

DesignKnowledge represents the reason and the development process of a product, existing in data resources, design cases, design tools, design processer and methods, standard specifications and talents, design experience, technology documents, and design results. Design knowledge may be organized and stored by KnowledgeTemplate, which is a structure of describing a certain kind of knowledge and is composed of attributes differing from other knowledge.

DesignDecision is the abstract class of the design decision, which is further specialized into selection decision, compromise decision, and hybrid decision. They are supported by decision templates respectively. DecisionTemplate is a formulation of a problem, through which all the template elements and the associated information are integrated. A problem can be formulated as multiple templates [51].

The main concepts and relationships of design knowledge and design decision model are shown in Figure 5.

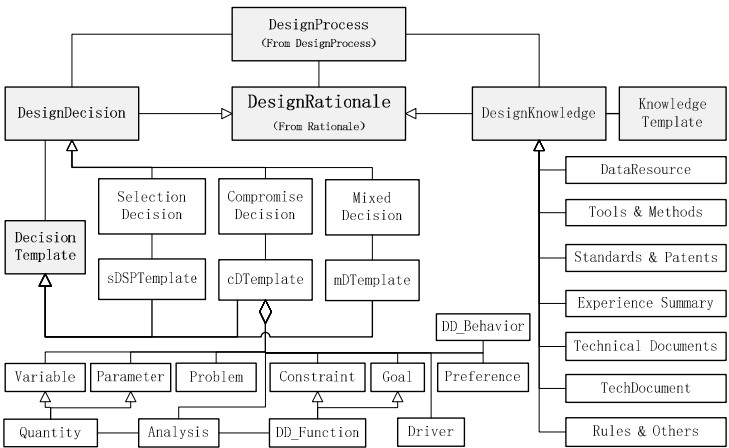

**Figure 5.** The main concepts and relationships of DesignRationale.

As shown in Figure 5, the concept DesignRationale is the specialization of Rationale of CPM. It can be further specialized into DesignKnowledge and DesignDecision. DesignKnowledge can be specialized into data resource, tools, methods, and so on, while DesignDecision can be further specialized into selection decision, compromise decision, and hybrid decision.

The expanded CPM model contains more concepts and semantic relationships, which need to be expressed in an effective way. Because of ontology's implementing interoperability, integrating the company applications, sharing information and reusing knowledge, etc., this article organizing the information of IMM based on ontology.

## 4. Representation of Intelligent Master Model Based on Ontology

The representation of IMM should not only satisfy the common understanding of product information for designers, but also support the computer readability and the interoperability between

different product development systems. The article chooses frame-based ontology method to represent the information of IMM for its advantages in standardized, computer readable, executable, and support-reasoning. A complete frame-ontology consists of concepts, relations among concepts, and consistency rules. Consistency rules are to keep the populated instances consistent to what they are defined to be. According to the extensions described above, the key concepts are identified and formally defined as classes in this section; the relations are identified and formally defined as slots, and the consistency rules are identified and formally defined. Finally, the IMM ontology (called OntoIMM) structure is presented.

### 4.1. Concept Identification

Domain concepts are often expressed in terms of vocabularies, researchers [1,15,28] have already identified many vocabularies of engineering product design field about CPM and its extension, such as Requirement, Function, Behavior, Form, Geometry, Material, Constraints, Parameter, and so on, which are reused in the IMM construct. In addition, these concepts DesignProcess, ProductControlStructure, DesignKnowledge, DesignDecision, ContextModel, and ViewModel, extended in the article based on CPM are also included, to enhance the semantic integrity and richness.

### 4.2. Relation Definition

Slots in frame ontology capture the semantic relationships among concepts. Typically, there are two types of slot: object slot and data slot. Object slot links a concept to an object data. The concept Function, for instance, links itself to the concept Behavior with an object slot. Data slot links a concept to a non-object data. The types of data slots include Integer, String, Float, URL, Symbol, and so on. For instance, every concept links itself to name, description and type with a data slot. The two slots of OntoIMM are defined in Tables 1 and 2, respectively and partly.

**Table 1.** Object slots of the OntoIMM (Partly).

| Slot Name | Definition | Type |
| --- | --- | --- |
| superclass/subclass_of | Link two concepts with super-class/subclass relationship | Instance |
| hasPart/is_part_of | Link two concepts with composition relationship | Instance |
| supports | Link DesignKnowledge and DesignDecision to DesignProcess | Instance |
| drives | Link a DesignKnowledge to a ProductControlStructure | Instance |
| updates | Link a ViewModel to a CoreProductModel+ | Instance |
| is_belong_to | Link a Parameter (parameter set) to a Structure | Instance |
| is_decided_on | Link a Parameter (parameter set) to a Function | Instance |
| needsMaterial | Link a Structure to a Material | Instance |
| has_restraint_for | Link a Function to a Structure<br>Link a Function to a Behavior | Instance |
| is_restrained_by | Link a Behavior to a Function | Instance |
| is_completed_by | Link a Structure to a DesignTask | Instance |
| is_comprised_by | Link a DesignProcess to a DesignGuide | Instance |
| hasVersion/version_of | Link a ProcessData to a Version(Versions) | Instance |
| hasMultilayerSkeleton | Link a PCS to a MultilayerSkeleton | Instance |
| hasControlParameter | Link a PCS to ControlParameters | Instance |
| hasDrivenKnowledge | Link a PCS to DrivenKnowledge | Instance |
| hasDecisionTemplate | Link a Decision to a DecisionTemplate | Instance |
| hasKnowledgeTemplate | Link a DesignKnowledge to a KnowledgeTemplates | Instance |

**Table 2.** Data slots of the OntoIMM (Partly).

| Slot Name | Definition | Type |
|---|---|---|
| IMMInfo | Information of IMM | String |
| name | Name of an instance | String |
| type | Type of an instance | String |
| information | Information of an instance | String |
| description | Description of an instance | String |
| processInfo | Information of a design process | String |
| PCSInfo | Information of PCS | String |
| skeletonInfo | Information of a multilayer skeleton | String |
| interfaceTemplateInfo | Information of a interface template | String |
| KTemplateInfo | Information of a knowledge template | String |
| knowledgeInfo | Information of a piece of knowledge | String |
| DTemplateInfo | Information of a decision template | String |
| functionInfo | Information of a Function | String |
| behaviorInfo | Information of a Behavior | String |
| structureInfo | Information of a Structure | String |
| materialInfo | Information of a Material | String |

## 4.3. Consistency Rules

IMM has more large-scale product data information and relationships. Keeping consistency to restrict the populated instances in the manner as they are defined to be is of critical importance. Inconsistency is prone to occur when if the conceptual model transforms, the IMM ontology needs to be modified, the analysis view model generates and updates, and so on. Thus, detecting the inconsistency and informing designers to address the inconsistency is very important. Rule-based reasoning is the method used for consistency checking in ontologies. In this paper, the rules for maintaining consistency in the OntoIMM are identified as shown in Table 3.

**Table 3.** Consistency rules of the OntoIMM.

| RuleNo | Rule Description |
|---|---|
| Rule1 | Each object and relationship has an Information attribute |
| Rule2 | Information is a container consisting of textual description slot, textual documentation string and properties slot |
| Rule3 | A properties slot that contains a set of attribute-value pairs stored as a string |
| Rule4 | Each object and relationship, except for the abstract and utility classes, has an attribute called type, the value of which is a string that acts as a symbolic classifier |
| Rule5 | Constraint is a specific shared property of a set of entities that must hold in all cases |
| Rule6 | There are associations existing between Specification and the Artifact that results from it |
| Rule7 | There are associations existing between a Flow and its source and destination Artifacts and its input and output Functions |
| Rule8 | There are associations existing between an Artifact and its Features |
| Rule9 | Function, Form and Behavior aggregate into Artifact |
| Rule10 | Function and Form aggregate into Feature |
| Rule11 | Geometry and Material aggregate into Form |
| Rule12 | Requirements aggregate into Specification |

SWRL is used to present the design rules. SWRL, which evolves from RuleML and conforms to W3C specifications, is a language that presents rules in semantic manner [52]. Based on OWL language, SWRL language integrates a variety of rules description methods to make up for the shortcomings of OWL in terms of rules description ability and semantic reasoning ability [53]. For example, the rule expression (?d hasCaliber ?12.7) means: Caliber is 12.7 mm; (BalanceShaft ?a) means: a is an instance of Balance Shaft.

The existing rule inference engine JESS (Java Expert Systems Shell) can only be embedded in protégé for application; it cannot be well integrated into other systems. However, the OntoIMM

proposed in this paper aims at supporting self-designed and developed integrated design system, and the inference machine based on JESS is difficult to meet the requirements of design system. Therefore, in the implementation of the integrated design system, the inference machine based on Jena was adopted because of its easy and flexible integration with any system. Jena is a Java development toolkit developed by HP Labs for application development in semantic web. Jena-based inference scheme first separates SWRL rules and ontology files from inference machine. Ontology is stored in the format of OWL file, while rules are stored separately in the format of rules file which is described in XML format and has fixed schema format definitions. An internal connection between OWL file and rules file is established. When performing the reasoning process, the owl ontology file is first analyzed in an inference machine, and then rules in rule file are analyzed and converted into a format recognized by the inference machine. Finally, the implicit relationship between the two elements is obtained by constructing the inference machine.

### 4.4. The Structure of Intelligent Master Model Ontology

Figure 6 shows the OntoIMM structure which is a network. In the structure, the classes are represented by the nodes. The solid-line arrows means that different classes are linked by object slots, and dashed-line arrows represents that the classes are linked to data by data slots. The Protégé tool, developed by Stanford University [54], is used to model the ontology.

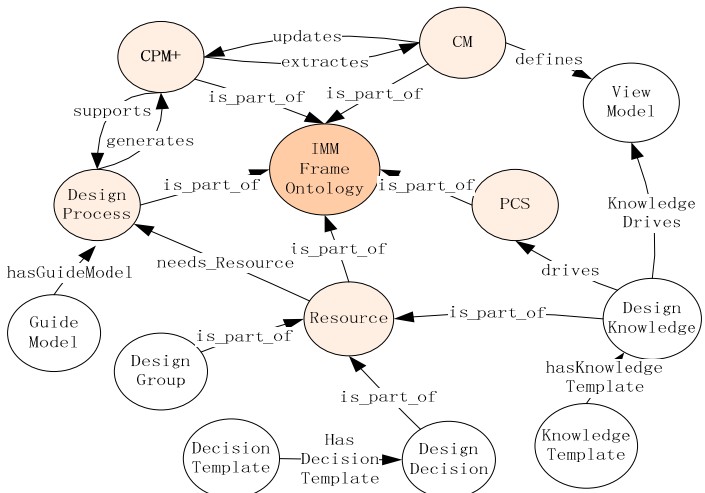

**Figure 6.** The structure of the OntoIMM (partly).

## 5. Case Study: Design and Analysis of Gun-Barrel

A Gun-barrel is the thick and hollow part (or component) that is composed of three characteristic surfaces: line chamber, chamber, and outside surface. It is one of the important parts (components) of automatic weapons and its main functions are giving bullets direction and initial velocity. The quality of barrel design and manufacturing directly influences the firing accuracy, life span, firing range, and weight. Barrel design should not only meet the requirements of structural dimensions and weight, but also meet sufficient strength and lifespan, so as to withstand the high temperature and high pressure of gunpowder gas and the friction and thermal shock between projectile and chamber.

Case-based variant design and adaptive design are two common strategies for designing small arms product, which can reduce time and cost consumption and improve the possibility of design by reusing the validated cases. For the design of barrel, new solutions are usually acquired based on the retrieval and modification of previous cases, especially in conceptual phases of product design. The variant design of barrel refers to similar case retrieval, parameter variant modifying, 3D model generating, strength checking, or FEA. Original design of barrel refers to interior ballistic design, rifling design, chamber throat design, materials selection, strength calculation, shape design and FEA, and

possibly thermal kinetics and modal analysis when necessary. Whether variant design or original design, the design of barrel will refer to several domains and each analysis is carried on based on the discipline-specific view model. Detailed information on its design and analysis process based on the OntoIMM method proposed is shown in Figure 7.

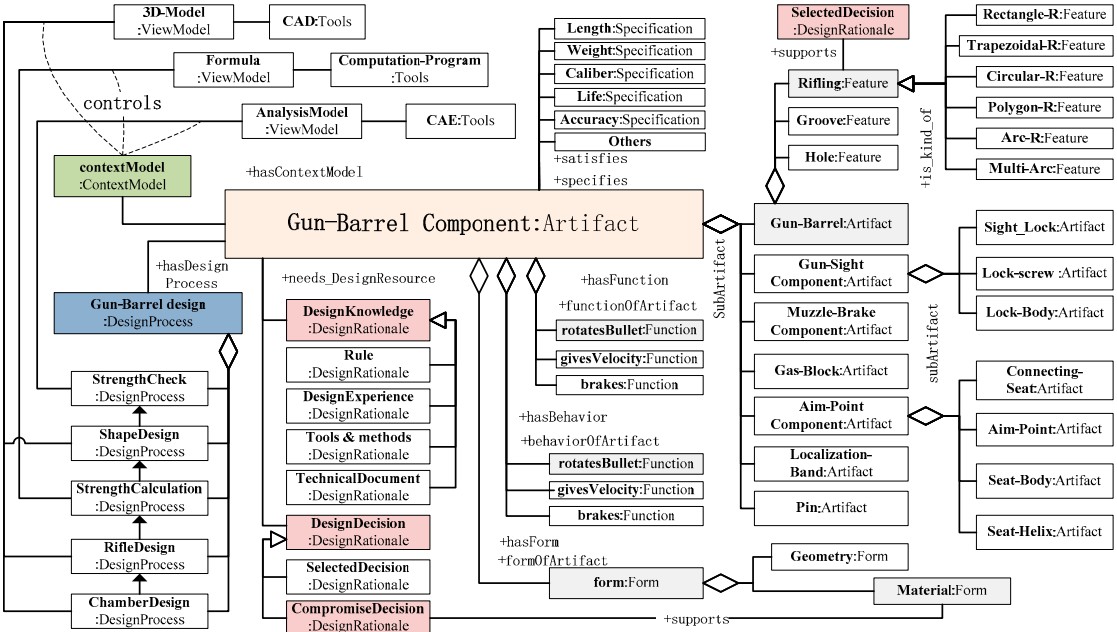

**Figure 7.** Detailed information for gun-barrel design analysis based on OntoIMM.

As Figure 7 depicts, the information concerning gun-barrel and its design process refers to requirement, function, behavior, component form, design process, and the corresponding analysis contexts such as 3D geometry modeling, strength calculating, and strength check based on FEA—the compromise decisions on strength calculating.

An intelligent design platform is developed based on J2EE technology framework. OntoIMM for product is modeled in protégé, and a function interface is developed to parse the ontology. The variant design process and original design process are defined, associated with their domain specific context and analysis view model. File-based integration method is adopted to encapsulate design and analysis relevant software tools (UG NX, Pro/E, MatLab) and other external tools (ANSYS) are called automatically through API interface. Design knowledge is managed and supplied when necessary. Taking the variant design of barrel for an example to validate the proposed method, the design analysis process is shown as follows.

### 5.1. Similar Case Acquiring

The acquisition of similar cases needs to calculate the similarity of product characteristics, which mainly includes the tactical and technical requirements for a firearms product. The concept Requirement is defined in OntoIMM and instantiated when parsed for reuse in intelligent design platform. Requirement is specialized into five types, including textual, numerical, interval, fuzzy, and containing, corresponding to similarity algorithms, respectively. Textual indicator refers to the descriptive indicators in this paper, such as fight task requirement; Numerical indicator is the quantitative description of an indicator, such as fighting rate of fire and theoretical rate of fire; Fuzzy indicator is expressed as membership function, such as material yield strength. The containing indicator is used to restrict the compatibility of products to specific objects or functions, for instance, a used bullet may contain several kinds of bullets. The selected requirement indexes and their values are shown in Table 4.

**Table 4.** The tactical and technical indexes.

| Selected Requirement | Type | Value |
|---|---|---|
| Caliber (mm) | numerical | 12.7 |
| Effective range (m) | numerical | 1500 |
| Whole weight limit(kg) | numerical | 25 |
| Whole length limit (mm) | numerical | 120 |
| Barrel length limit (mm) | numerical | 600 |
| Initial velocity limit (m/s) | numerical | 750 |
| Theoretical rate of fire (round /min) | numerical | 800 |
| Fighting rate of fire (round /min) | numerical | 300 |
| Used bullet | containing | Armor-piercing incendiary, type 54, 12.7 mm |
| Fight task | textual | Flexible operation, detachable transport, high reliability and low failure rate |
| Material | textual | Easy access, enough strength and low cost |
| Maintenance | textual | Standards conformance and good maintainability |

The similarity algorithms code integrated are called to comprehensively calculate the similarity of requirement characteristics. The comprehensively similarity of a design case can be expressed as

$$\text{Sim}(X,Y) = \omega_t \text{Sim}_t + \omega_n \text{Sim}_n + \omega_i \text{Sim}_i + \omega_f \text{Sim}_f + \omega_c \text{Sim}_c, \tag{1}$$

where $\omega_t$, $\omega_n$, $\omega_i$, $\omega_f$, and $\omega_c$ are the weight of textual, numerical, interval, fuzzy and containing indexes, respectively, and their values range from 0 to 1. $\text{Sim}_t$, $\text{Sim}_n$, $\text{Sim}_i$, $\text{Sim}_f$, and $\text{Sim}_c$ are the similarity of textual, numerical, interval, fuzzy, and containing indexes. Taking numerical indexes as an example, the calculation formula of similarity is represented by

$$\text{SIM}_n(X, Y) = 1 - \frac{|X_{val} - Y_{val}|}{\text{MAX}_{val} - \text{MIN}_{val}}, \tag{2}$$

where X and Y are the same numerical index; $\text{MAX}_{val}$ and $\text{MIN}_{val}$ are the upper and lower bounds of X and Y, respectively; and $X_{val}$ and $Y_{val}$ are the specific values of X and Y, respectively.

Several similar cases will be obtained, and the specific value and similarity of each index of selected similar case will be shown on the system interface to help engineers view and select, as shown in Figure 8.

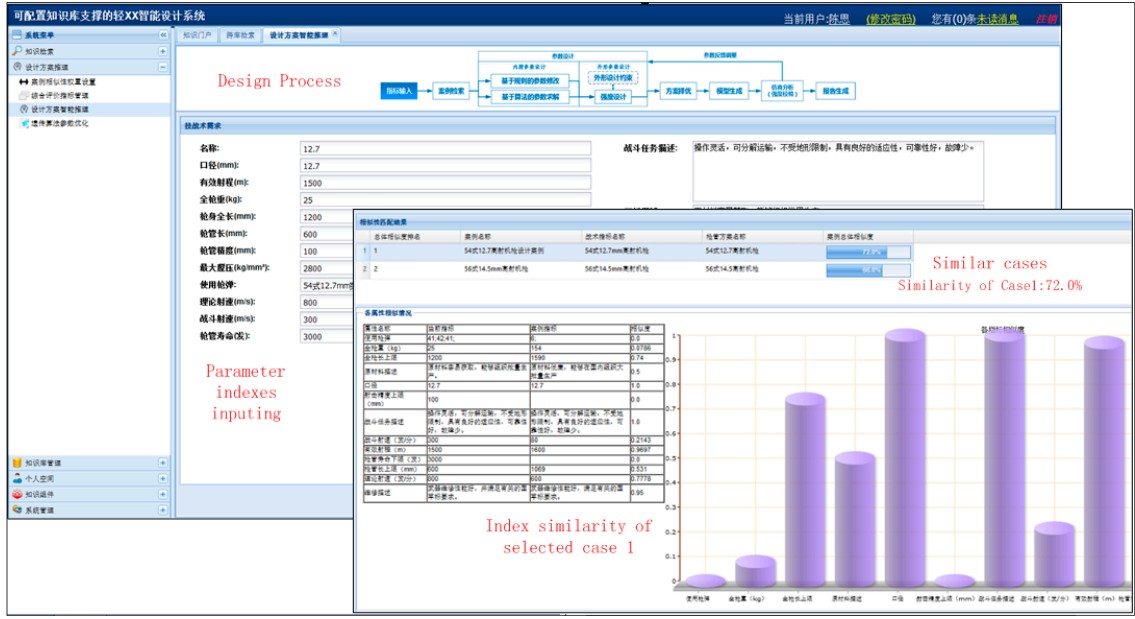

**Figure 8.** Similar case acquiring.

As shown in Table 4, the comprehensive similarity of case is displayed on the user interface, and the index value will be displayed according to the selected case by model mapping. The input values of selected requirements will initial fulfill the barrel design information, which will be used and updated at later design stages.

## 5.2. Parameter Variant Modifying

As the initial scheme of the current design, the previous design examples do not meet the existing design requirements inevitably. Thus, the initial schemes need to be further modified to meet requirements of new design problems. Designers usually make engineering change decision based on personal experience and relevant design knowledge. The design system provides automatically relevant design knowledge for the modification interactively, as shown in Figure 9.

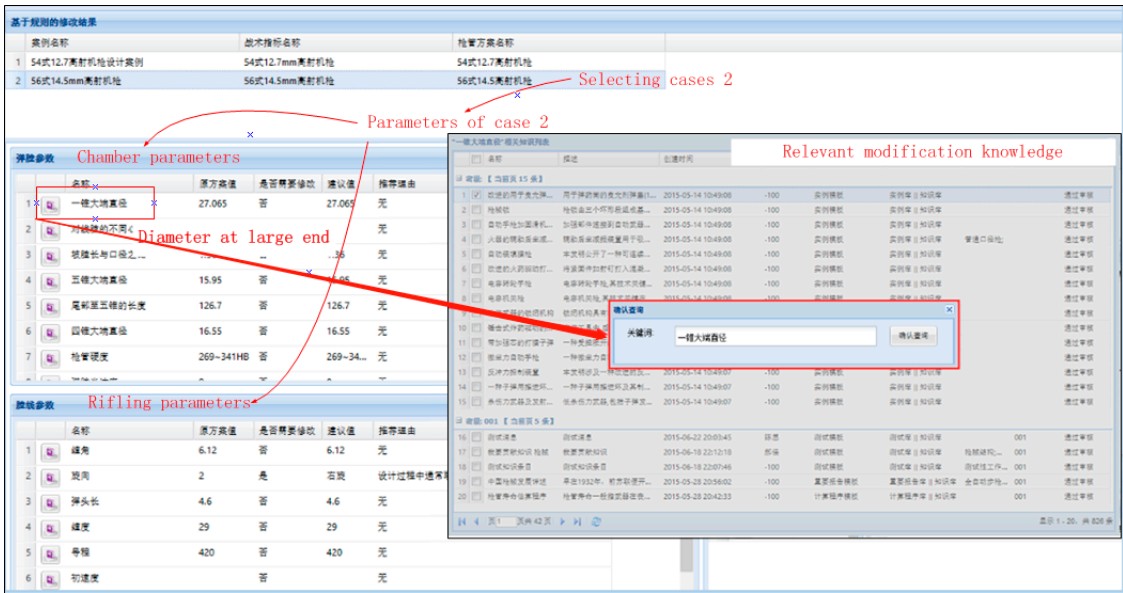

**Figure 9.** Parameter modifying based on design knowledge.

### 5.2.1. Rule-Based Parameter Modifying

Designers make the parameter modification by personal experience or with the help of rule knowledge. For instance, the knowledge rule "if the caliber is less than or equal to 9 mm, then the number of rifling is usually 4" will be provided to designer when the number of rifling needs to be modified.

### 5.2.2. Compromise Decision-Based Parameter Modifying

For the determination of the wall thickness which forms the theoretical shape of barrel, strength calculation is needed to be conducted based on compromise decision. It refers to gaining the relationship between chamber pressure, bullet velocity and bullet range, time based on interior ballistic calculation with a certain shoot condition of loading and bullet. In this step, chamber pressure characteristics are acquired to draw the calculated chamber pressure curve (P–L curve) with MATLAB, and wall thicknesses are determined based on the view data (internal diameters of three main sections) with calculation code. According to the compromise decision model described in Section 3.6, the information needed for design decision calculation is automatically extracted from the existing product information, as shown in Table 5.

**Table 5.** The compromise decision template for strength calculating.

| DTemplateInfo | Name | | Type | Object | OtherInfo |
|---|---|---|---|---|---|
| | Strength calculation | | compromise decision | Barrel | . . . |
| | **Name** | **Type** | **Unit** | **Value** | **Input/Output** |
| Parameter | $\sigma s$ | numerical | N/mm$^2$ | 50 | Input |
| | BL | numerical | mm | 600 | Input |
| | P–L curve | Matlab file | Unit | P–L.mat | Input |
| | r1 | numerical | mm | 12.7 | Input |
| | FG | numerical | — | FG $=\omega_1*r_2 +\omega_2*$W | Input |
| | **Name** | **Type** | **Unit** | **Value** | **Behavior** |
| Variable | r2C | numerical | mm | Output | Output |
| | r2MBP | numerical | mm | Output | Output |
| | r2M | numerical | mm | Output | Output |
| | **Name** | **Type** | **Constraint description** | | |
| Constraint | n | hard | 0.9–1 in chamber section<br>1.2–1.3 in MBP section<br>3–5 in muzzle section | | |
| | **Name** | **Type** | **Description** | **Weight** | **Formula** |
| Goal | r2 | Max | Max r2 | $\omega1$ | r2 = f (r1,$\sigma$s, n, p) |
| | W | Min | Min weight | $\omega2$ | W = f (r2, r1, BL, ST, $\rho$) |
| Analysis | Algorithm of multi-objective analysis, Process of barrel strength calculation | | | | |
| Driver | Matlab for P–L curve, Code for barrel strength calculation | | | | |
| Preference | Design rule 1, Design rule 2, . . . | | | | |
| History | Previous experience | | | | |
| Response | Optional parameters | | | | |

Table 5 describes the decision elements of the strength calculation. The nomenclatures in the template are defined in Table 6.

**Table 6.** Nomenclatures in the template.

| Nomenclatures | Description |
|---|---|
| BL | Barrel length (mm) |
| $r_{2C}$ | External diameter in section of chamber (mm) |
| $r_{2MBP}$ | External diameter in section of MBP (mm) |
| $r_{2M}$ | External diameter in section of muzzle (mm) |
| FG | Goal function |
| n | Safety Factor |
| W | Barrel weight (kg) |
| $\omega_1$ | Weight associated with the $r_2$ goal |
| $\omega_2$ | Weight associated with the W goal |
| $r_1$ | Internal diameter (mm) |
| p | Pressure of the gunpowder gas in the chamber (kpa) |
| $\rho$ | Material density (g/cm$^3$) |
| $\sigma_s$ | Material yield limit (N/mm$^2$) |
| ST | The shape type of the rifling cross section |
| P–L | Calculated bore pressure curve |

The compromise decision code for strength calculation is automatically called to visually show the optional parameters (the wall thickness of critical locations) and variant sensitivity, and the shape parameters are obtained.

### 5.3. 3D Model Generating

When an optimal solution is selected, the geometry characteristics are extracted and injected to barrel PCS, which drives CAD tools (UG NX) to generate 3D model. The run process and results of selected barrel, and a version of the result are shown in Figure 10. If the PCS is called again with modifications, another version will be acquired accordingly.

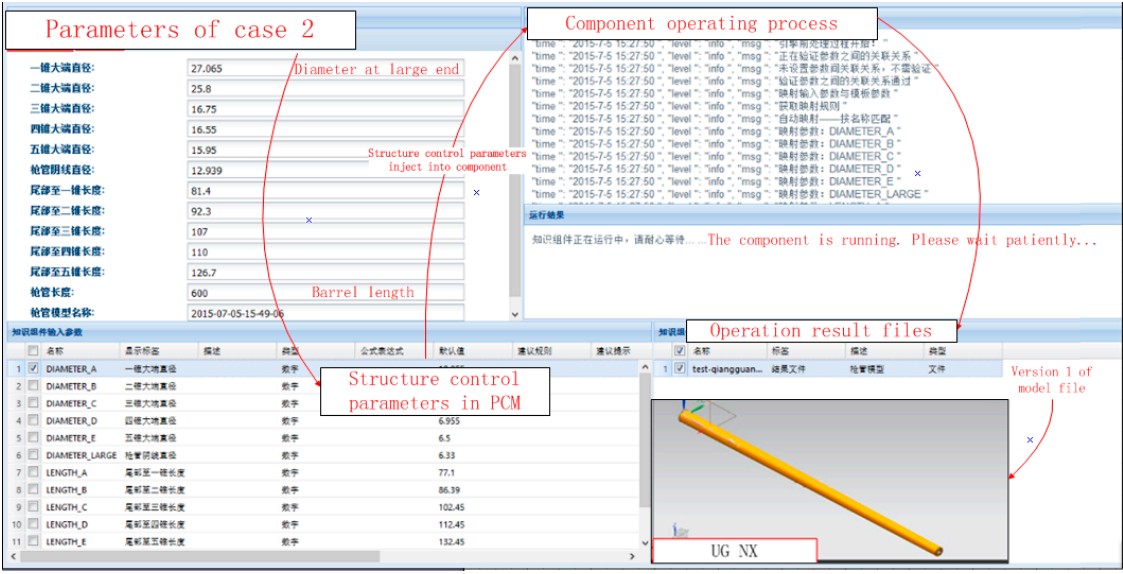

**Figure 10.** 3D model generating process.

### 5.4. Strength Checking or FEA

After generating the 3D model, strength checking is needed to verify that the whole model to meet the strength requirement, as shown in Figure 11.

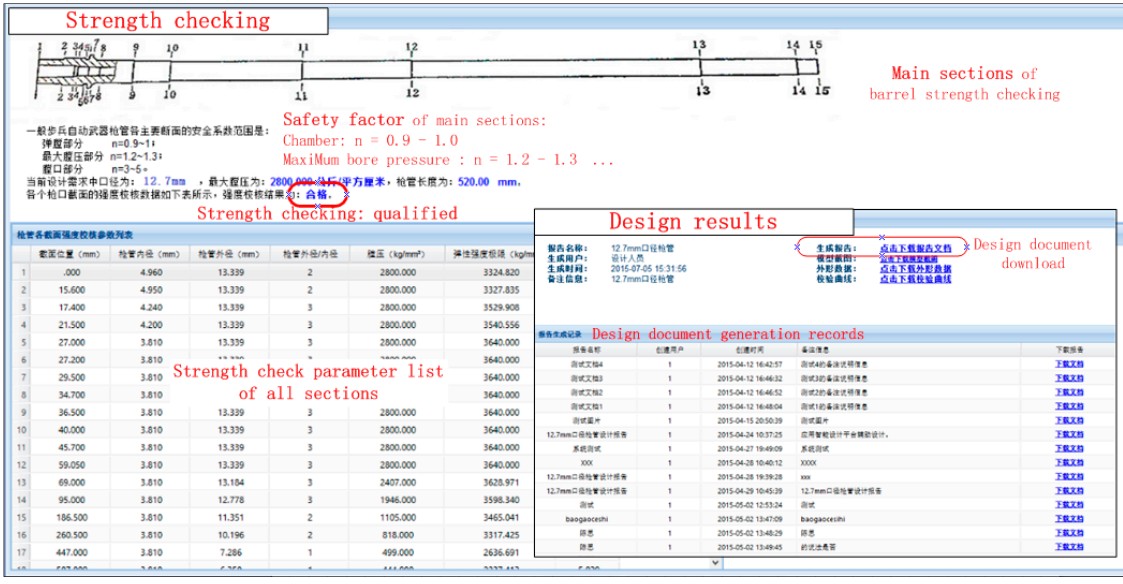

**Figure 11.** Strength checking and design documents.

Strength checking requires the knowledge component to invoke the MATLAB to calculate the safety factor n. The formula is defined as follows

$$n = \frac{1}{p} \cdot \sigma_s \cdot \frac{3(a^2 - 1)}{2(2a^2 + 1)},$$ (3)

where p, n, and $\sigma_s$ are defined in Table 6, and a is the ratio of $r_2$ to $r_1$.

As depicted in Figure 11, the checking result is labeled "Qualified". For single part, it is usually adequate to use experience formula, such as Equation (3), for strength checking. However, FEA may be needed for more accurate checking or for barrel component. FEA tools, such as ANSYS which encapsulate the knowledge about pre- and postprocessing and running orders, are integrated with the OntoIMM.

## 6. Conclusions

At present, there is a lack of systematic research on IMM. The author creatively defines the intelligent master model from the view that IMM is the carrier of product information and knowledge, the integration center, and the service agent of design process. On this basis, systematically, studies on IMM from aspects of definition, information modeling, design knowledge modeling, and model transformation have been carried out. The main work of this paper is to establish information organizing mechanism, which is one of the two key issues of IMM. The original intention is based on two confirmed theoretical foundations: (1) NIST CPM can serve as the information organizing mechanism for product master model and (2) intelligent master model is an enhancement of product master model based on KBE. When CPM is adopted to act as the information organizing principle for IMM, there exist two shortcomings: lack of the representation of complete information and knowledge and lack of semantic richness. It limits the ability of integration model to support design process and system. Therefore, based on design process analysis in introduction and related work review, we define a multiaspect extension to CPM and then construct an ontology to represent the information and design knowledge. First, in order to integrate the detailed description of design process, such as design task, design activity, process elements, process states, and basis attributes, the class ProcessInformation of CPM is extended to design process model. Secondly, the design rationale model expands the concept connotation of Rationale of CPM to organize design knowledge and decision information. In addition, another two models—product control structure model and context model—are added to represent the product structure control information and context information. An ontology is constructed to represent related concepts, relationships, and rules. Finally, the effectiveness of the proposed method is verified by an example of gun barrel design and analysis process.

However, the research mainly concentrates on constructing the information organizing mechanism for intelligent master model, which is the basic and key issue of IMM. The future and further work may be in model implementation and management, including acquiring and serving multilayer design knowledge (especially design rules), model transformation, and maintaining consistency in model transformation and modification.

**Author Contributions:** C.Y. and F.Z. conceived and designed the method; Y.Y., S.I.B., and W.L. analyzed the data and contributed to case study; C.Y. wrote the article.

**Funding:** This research received no external funding.

**Acknowledgments:** This work is supported by National key research and development program (2018YFB1701802) and National ministry project (JCKY2017207A001). The authors thank anonymous reviewers for their helpful suggestions in this study.

**Conflicts of Interest:** The authors declare no conflict of interest.

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
