# Peer review of "OntoIMM: An Ontology for Product Intelligent Master Model"

_applsci, doi:10.3390/app9122553_

Round 1

Reviewer 1 Report

Maybe the example of a gun / gun-barrel is not ideal !?

line 13: KBE, definition missing
line 21: ProcessInformation, definition missing or process information
line 21: Rationale definition missing or rational
line 35, 273, 277, 284, 286: con-trolles, please check entire document for "-"
line 85+193+194: Artifact definiton missing or artifact, also see line 193 and line 194
line 99: to represent(s)
line 99-106: whole passage has been copied, check line 20-27 !!! NOT acceptable!!!
line 187: space missing process_[28] please check entire document
line 205, 332: space missing [29,_30] please check entire document
line 197: geometry and material
line 199: information, process information and rationale
line 212, 214 ff.: please check time of references: present/presented describe/described
line 284: spelling
line 427: space missing templates_[
line 436: hard section break, between 3.6 and 4, please add binding text
line 448: space missing [1,_]
line 475: please introduce SWRL or add an reference
line 473 and table 3: Please explain, these rules are for maintaining not coded in IMM or Protege; right?
line 506: space Figure_7
line 515: tool UG unknown, maybe references for 3/4 tools?
line 523 and Table 4: Explanation of type fuzzy required. Values in table 4 are not fuzzy.
line 539 and 552: Figure 9 named twice, please check number.
line 538: Quality of figure 9/1 not acceptable.
line 551: Quality of figure 9/2 not acceptable.
line Table5 : Question: In my understanding this tabe is a description of the gun-barrel and parameter.
Why is it not represented as an ontology? e.g. as RDF (for the ontology) and RDF data (for instances and values)?
line 582 and 587: Quality of figure 10 and 11 not acceptable.
line 613: consistency rules are not forced! Check line 473 and table 3, as well as comment.
These rules are for maintaining not coded in IMM or Protege.
line 606-611 : whole passage has been copied, check line 20-27 and again line 99-106: !!! NOT acceptable!!!

Author Response

Dear reviewer,

Thank you very much for your suggestions for our manuscript (Manuscript Number: applsci-527246; Title: OntoIMM: An Ontology for Product Intelligent Master Model.

We have read your comments very carefully. As for your advices, we checked out our article carefully and answered all questions one by one as follows. The part of modification has been highlighted with red font in the text.

1. line 13: KBE, definition missing.

Reply: Thank you for your advice. The definition of KBE is added in Line 14.

2. line 21: ProcessInformation, definition missing or process information

Reply: As for this advice, the definition of ProcessInformation was highlighted in Related works section. Please refer to Line 187.

3. line 21: Rationale definition missing or rational

Reply: Thanks for your comment. The definition of Rationale was highlighted in Related works section. Please refer to Line 189 and 190.

4. line 35, 273, 277, 284, 286: con-trolles, please check entire document for "-"

Reply: Grateful for your suggestion. This whole manuscript has been examined carefully and revised.

5. line 85+193+194: Artifact definiton missing or artifact, also see line 193 and line 194

Reply: Based on the advice of the reviewer, the definition was added in Related works section. Please refer to Line 175-179.

6. line 99: to represent(s)

Reply: According to your advice, it has been corrected in Line 94.

7. line 99-106: whole passage has been copied, check line 20-27 !!! NOT acceptable!!!

Reply: Thank you for your valuable advice. These sentences have been rewritten as “Firstly, four submodels are added to CPM to comprehensively describe the reason, process and result information and knowledge about product. Secondly, the concepts, relationships and consistency rules of IMM information structure are modeled in a ontology.” Please refer to Lines 94-97.

8. line 187: space missing process_[28] please check entire document

Reply: According to your comment, spaces have been added in this article where necessary, and the full text has been checked and modified.

9. line 205, 332: space missing [29,_30] please check entire document

Reply: Thanks for your advice. Spaces have been added to this article where necessary, and the full text has been checked and modified.

10. line 197: geometry and material

Reply: Grateful for your suggestion. Geometry and Material are two object classes in CPM to describe the spatial description and the internal composition of a artifact, respectively. In order to distinguish them, concepts or classes in this article are represented in uppercase and italics. Please refer to Line 184.

11. line 199: information, process information and rationale

Reply: As for this suggestion, Information, ProcessInformation and Rationale are three utility classes of CPM. In order to distinguish them, concepts or classes in this article are represented in uppercase and italics. Please refer to Line186.

12. line 212, 214 ff.: please check time of references: present/presented describe/described

Reply: Thanks for your comments. We checked out our manuscript carefully and revised language of the whole article.

13. line 284: spelling

Reply: Your suggestions were very much appreciated. Those mistakes have been corrected and we have carefully checked the whole manuscript to confirm that there are no other similar errors in the paper.

14. line 427: space missing templates_[

Reply: Thank you for your advice. Pertinent space missing was modified in manuscript.

15. line 436: hard section break, between 3.6 and 4, please add binding text

Reply: As for this suggestion, binding text has been added in Line 420-423.

16. line 448: space missing [1,_]

Reply: According to your comment, pertinent space missing was modified in manuscript.

17. line 475: please introduce SWRL or add an reference

Reply: Grateful for your suggestion. In this paper, SWRL has been briefly introduced in Line 459-464 and two literatures [54] and [55] are added.

18. line 473 and table 3: Please explain, these rules are for maintaining not coded in IMM or Protege; right?

Reply: Your suggestions were very much appreciated. In order to make it more clear, we have made relevant explanations in Line 466-479 of the manuscript.

19. line 506: space Figure_7

Reply: Based on your advice, pertinent space missing was modified in manuscript.

20. line 515: tool UG unknown, maybe references for 3/4 tools?

Reply: According to your advice, “UG” has been modified to “UG NX” in Line 517.

21. line 523 and Table 4: Explanation of type fuzzy required. Values in table 4 are not fuzzy.

Reply: Grateful for your suggestion. The type of Effective range, Whole weight limit, Whole length limit, Barrel length limit, Initial velocity limit, Theoretical rate of fire, Fight rate of fire has been modified to be numerical, and explanations of the indicators has been added in Line 526-531.

22. line 539 and 552: Figure 9 named twice, please check number.

Reply: According to your advice, The figure number of line 546 has been corrected to Figure 8.

23,24. line 538: Quality of figure 9 not acceptable.

Reply: Your suggestions were very much appreciated. A higher quality figure was used for the substitution in Line546.

25. line Table5 : Question: In my understanding this table is a description of the gun-barrel and parameter. Why is it not represented as an ontology? e.g. as RDF (for the ontology) and RDF data (for instances and values)?

Reply: Thank you very much for your valuable advices. Authors of reference [53] and this paper are on the same team working together to develop an integrated design system. Reference [53] mainly focuses on the construction of template of compromise decision support problem based on ontology. The intelligent master model proposed in this paper is to provide a base model of this design system, so as to integrate various information modules of the system and realize information exchange and knowledge sharing. Therefore, the decision template modeling is not further elaborated in this paper.

26. line 582 and 587: Quality of figure 10 and 11 not acceptable.

Reply: According to your suggestions. Higher quality figures were used for the substitution in Line586 and 591.

27. line 613: consistency rules are not forced! Check line 473 and table 3, as well as comment. These rules are for maintaining not coded in IMM or Protege.

Reply: Thanks for you advice. In order to make it more clear, we have made relevant explanations in Line 466-479 of the manuscript.

28. line 606-611 : whole passage has been copied, check line 20-27 and again line 99-106: !!! NOT acceptable!!!

Reply: Thanks for your advice. These sentences have been rewritten in Lines 611-621.

In all, we found the reviewer’s comments are quite helpful to improve our paper, and the paper has been revised point by point. In addition, we also make some other modifications to improve the paper, and some language descriptions are improved. We hope the present version is more acceptable for publication.

We must extend our sincere gratitude again for your great comments and suggestions.

Best regards.

Yours sincerely,

Fa-ping Zhang

zfp_new@163.com

Reviewer 2 Report

The paper is focused on presenting an ontology for product intelligent master model (IMM). This paper aims at addressing the information organizing mechanism. According to the Authors, the effectiveness of proposed method was verified on base of the example of design and analysis of gun-barrel. The paper is well-positioned to the journal aim and scope.

The structure of the paper is clear. The foundations of IMM and CPM were introduced/ investigated by Authors as well as motivation to use the ontology for modelling IMM caused by its support on a closed-world assumption. However, there are shortcomings in this paper:

1.     Why the Authors used the old version of Protégé software (3.5.)?  - based on the reference numbered as [54]. It should be updated using up to date software

2.     As the Authors adapted IMM model to construct the ontology, what is the Authors contribution to this model? This process should be clearly described.

3.     Main idea of ontology building is knowledge sharing. I have a question if it is possible to provide public available version of authors’ ontology? It is easy to realize even using webprotege software

English language:

Minor typos:

In this paper there are some typography errors – the Authors should avoid the single letters at the end of lines, the paper is plenty of that type of mistakes.

pp. 6, line 266: “….depen-dency…” – please check the syntax.

Author Response

Dear reviewer,

Thank you very much for your suggestions for our manuscript (Manuscript Number: applsci-527246; Title: OntoIMM: An Ontology for Product Intelligent Master Model. 

We have read your comments very carefully. As for your advices, we checked out our article carefully and answered all questions one by one as follows. The part of modification has been highlighted with red font in the text.

1. Why the Authors used the old version of Protégé software (3.5.)?  - based on the reference numbered as [54]. It should be updated using up to date software

Reply: Your suggestions were very much appreciated. This paper comes from the support of basic scientific research projects. In the project at that time, Protege software (3.5.) was used to construct ontology, so the article also adopted the version of that time. However, newer version of protégé has been used in new basic research projects.

2. As the Authors adapted IMM model to construct the ontology, what is the Authors contribution to this model? This process should be clearly described.

Reply: Thanks for your good advice. In order to make it more clearly, we have made relevant explanations in Line 602-607 of the manuscript.

3. Main idea of ontology building is knowledge sharing. I have a question if it is possible to provide public available version of authors’ ontology? It is easy to realize even using webprotege software.

Reply: Thank you for your valuable comment. Ontology construction is not so difficult in protege. After all, the difficulty of ontology construction lies in accurate and comprehensive sorting and defining concepts and relationships. Based on the basic scientific research project, the author participated in the construction of the tank design ontology and the gun design ontology, but it was not disclosed to the public due to some special reasons. The author of this paper is trying to do ontology construction in some general fields, such as industrial Internet field, and will write relevant research papers or reports for publication.

4. English language:

Minor typos:

In this paper there are some typography errors – the Authors should avoid the single letters at the end of lines, the paper is plenty of that type of mistakes. 

pp. 6, line 266: “….depen-dency…” – please check the syntax.

Reply: Based on your advice, the whole paper has been checked and revised to avoid the single letters at the end of lines. 

The sentence “….depen-dency…” has been rewritten in Line 254-255.

Furthermore, we checked out our manuscript carefully and revised other mistakes.

In all, we found the reviewer’s comments are quite helpful to improve our paper, and the paper has been revised point by point. In addition, we also make some other modifications to improve the paper, and some language descriptions are improved. We hope the present version is more acceptable for publication.

We must extend our sincere gratitude again for your great comments and suggestions.

Best regards.

Yours sincerely,

Fa-ping Zhang

zfp_new@163.com
